# VIBIDSAMPLER: ENHANCING VIDEO INTERPOLATION USING BIDIRECTIONAL DIFFUSION SAMPLER

**Serin Yang**[1*], **Taesung Kwon**[2*], **Jong Chul Ye**[1]
[1]Kim Jaechul Graduate School of AI, KAIST    [2]Dept. of Bio & Brain Engineering, KAIST
{yangsr, star.kwon, jong.ye}@kaist.ac.kr

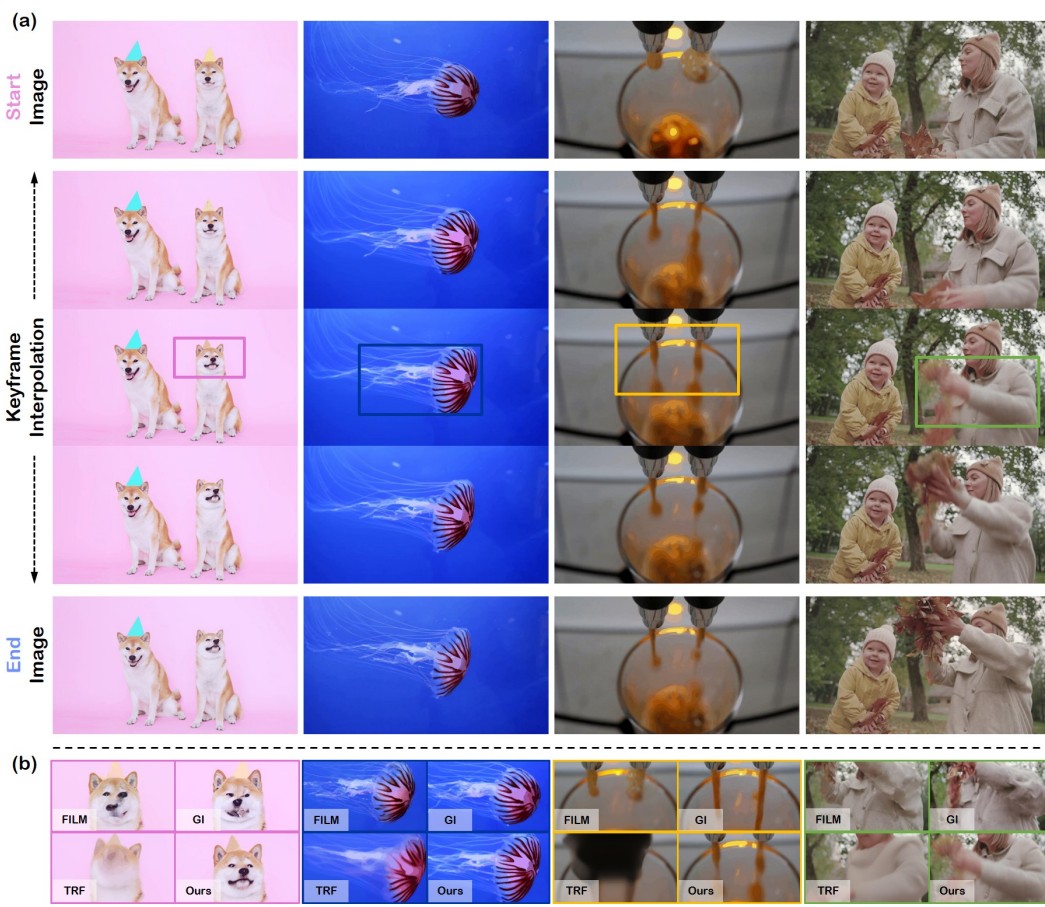

Figure 1: **Keyframe interpolation results using our ViBiDSampler.** (a) The images in the first and last rows are keyframes, and the intermediate frames are generated using ViBiDSampler. (b) A comparison of results with three baseline methods—FILM, TRF, and Generative Inbetweening (GI)—demonstrates that these baselines exhibit artifacts or unnatural appearances. In contrast, our method generates clear and realistic frames.

## ABSTRACT

Recent progress in large-scale text-to-video (T2V) and image-to-video (I2V) diffusion models has greatly enhanced video generation, especially in terms of keyframe interpolation. However, current image-to-video diffusion models, while powerful in generating videos from a single conditioning frame, need adaptation for two-frame (start & end) conditioned generation, which is essential for effective bounded interpolation. Unfortunately, existing approaches that fuse temporally forward and backward paths in parallel often suffer from off-manifold issues, leading to artifacts or requiring multiple iterative re-noising steps. In this work, we introduce a novel, bidirectional sampling strategy to address these off-manifold issues without requiring extensive re-noising or fine-tuning. Our method

---

*Equal contribution.

employs sequential sampling along both forward and backward paths, conditioned on the start and end frames, respectively, ensuring more coherent and on-manifold generation of intermediate frames. Additionally, we incorporate advanced guidance techniques, CFG++ and DDS, to further enhance the interpolation process. By integrating these, our method achieves state-of-the-art performance, efficiently generating high-quality, smooth videos between keyframes. On a single 3090 GPU, our method can interpolate 25 frames at $1024{\times}576$ resolution in just 195 seconds, establishing it as a leading solution for keyframe interpolation. Project page: https://vibidsampler.github.io/

# 1 INTRODUCTION

Recent advancements in large-scale text-to-video (T2V) and image-to-video (I2V) diffusion models (Blattmann et al., 2023a;b; Wu et al., 2023; Xing et al., 2023; Bar-Tal et al., 2024) have made it possible to generate high-quality videos that closely match a given text or image conditions. Various efforts have been made to leverage the powerful generative capabilities of these video diffusion models, especially in the context of keyframe interpolation, to improve perceptual quality significantly. Specifically, diffusion-based keyframe interpolation (Voleti et al., 2022; Danier et al., 2024; Huang et al., 2024; Feng et al., 2024; Wang et al., 2024) focuses on generating intermediate frames between two keyframes, aiming to create smooth and natural motion dynamics while preserving the keyframes' visual fidelity and appearance. Image-to-video diffusion models are particularly well-suited for this task because they are designed to maintain the visual quality and consistency of the initial conditioning frame.

While image-to-video diffusion models are designed for start-frame conditioned video generation, they need to be adapted for start and end frame conditioned video generation for keyframe interpolation. One line of works (Feng et al., 2024; Wang et al., 2024) addresses this issue by introducing a new sampling strategy that fuses the intermediate samples of the temporally forward path, conditioned on the start frame, and the temporally backward path, conditioned on the end frame. The fusing strategy ensures smooth and coherent frame generation in-between two keyframes using image-to-video diffusion models in a training-free (Feng et al., 2024) or a lightweight fine-tuning manner (Wang et al., 2024).

In the geometric view of diffusion models (Chung et al., 2022), the sampling process is typically described as iterative transitions $\mathcal{M}_t \rightarrow \mathcal{M}_{t-1}, t = T, \cdots, 1$, moving from the noisy manifold $\mathcal{M}_T$ to the clean manifold $\mathcal{M}_0$. From this perspective, fusing two intermediate sample points through linear interpolation on a noisy manifold can lead to an undesirable off-manifold issue, where the generated samples deviate from the learned data distribution. TRF (Feng et al., 2024) reported that this fusion strategy often results in undesired artifacts. To address these discrepancies, they apply multiple rounds of re-noising and denoising to the fused samples, which may help correct the off-manifold deviations.

Unlike the prior works, here we introduce a simple yet effective sampling strategy to address off-manifold issues. Specifically, at timestep $t$, we first denoise $\boldsymbol{x}_t$ to obtain $\boldsymbol{x}_{t-1}$ along the temporally forward path, conditioned on the start frame ($I_{\text{start}}$). Then, we re-noise $\boldsymbol{x}_{t-1}$ back to $\boldsymbol{x}_t$ using stochastic noise. After that, we denoise $\boldsymbol{x}'_t$ to get $\boldsymbol{x}'_{t-1}$ along the temporally backward path, conditioned on the end frame ($I_{\text{end}}$), where the $'$ notation indicates that the sample has been flipped along the time dimension. Unlike the fusing strategy, which computes two conditioned outputs in parallel and then fuses them, our *bidirectional diffusion sampling* strategy samples two conditioned outputs sequentially, which mitigates the off-manifold issue.

Furthermore, we incorporate advanced on-manifold guidance techniques to produce more reliable interpolation results. First, we employ the recently proposed CFG++ (Chung et al., 2024), which addresses the off-manifold issues inherent in Classifier-Free Guidance (CFG) (Ho & Salimans, 2021). Second, we incorporate DDS guidance (Chung et al., 2023) to ensure proper alignment of the last frame of the generated samples with the given frames, as the ground-truth start and end frames are already provided. By combining bidirectional sampling with these guidance techniques, our method achieves stable, state-of-the-art keyframe interpolation performance without requiring fine-tuning or multiple re-noising steps. Thanks to its efficient sampling strategy, our method can interpolate between two keyframes to generate a 25-frame video at $1024{\times}576$ resolution in just 195 seconds on a single 3090 GPU. Since our method is designed for high-quality and *vivid* video keyframe interpola-

tion using bidirectional diffusion sampling, we refer to it as **V**ideo **I**nterpolation using **BI**directional **D**iffusion (ViBiD) Sampler.

## 2  RELATED WORKS

**Video interpolation.** Video interpolation is a task that generates the intermediate frames based on two bounding frames. Conventional interpolation methods have utilized convolutional neural networks (Kong et al., 2022; Li et al., 2023; Lu et al., 2022; Huang et al., 2022; Zhang et al., 2023b; Reda et al., 2019), which are typically trained in a supervised manner to estimate the optical flows for synthesizing an intermediate frame. However, they primarily focus on minimizing $L_1$ or $L_2$ distances between the output and target frames, emphasizing high PSNR values at the expense of perceptual quality. Furthermore, the train datasets generally consist of high frame rate videos, limiting the model's ability to learn extreme motion effectively.

**Diffusion-based methods and time reversal sampling.** Diffusion-based methods have been proposed (Danier et al., 2024; Huang et al., 2024; Voleti et al., 2022) to leverage the generative priors of diffusion models to produce high-quality perceptual intermediate frames. Although these methods demonstrate improved perceptual performance, they still struggle with interpolating frames that contain significant motion. However, video keyframe interpolation methods that build on the robust performance of video diffusion models have been more successful in handling ambiguous and nonlinear motion (Xing et al., 2023; Jain et al., 2024), largely due to the incorporation of the temporal attention layers in these models (Blattmann et al., 2023a; Ho et al., 2022; Chen et al., 2023; Zhang et al., 2023a).

Recent advancements in video diffusion models, particularly for image-to-video diffusion, have introduced new sampling techniques that leverage temporal and perceptual priors. These techniques reverse video frames in parallel during inference and fuse bidirectional motion from both the temporally forward and backward directions. TRF (Feng et al., 2024) proposed a method that combines forward and backward denoising processes, each conditioned on the start and end frames. Similarly, Generative Inbetweening (Wang et al., 2024) introduced a method that extracts temporal self-attention maps and rotates them to sample reversed frames, enhancing video quality by fine-tuning diffusion models for reversed motion. However, these methods rely on a fusion strategy that often results in an off-manifold issue. Moreover, although methods such as multiple noise injections and model fine-tuning have been employed to address these challenges, they continue to exhibit off-manifold issues and substantially increase computational costs. In contrast, we introduce a simple yet effective sampling strategy that eliminates the need for multiple noise injections or model fine-tuning.

## 3  VIDEO INTERPOLATION USING BIDIRECTIONAL DIFFUSION

Although our method is applicable to general video diffusion models, we employ Stable Video Diffusion (SVD) (Blattmann et al., 2023a) as a proof of concept in this paper. By introducing SVD, we aim to provide a clearer understanding of our approach. SVD is a latent video diffusion model employed in EDM-framework (Karras et al., 2022) with micro-conditioning (Podell et al., 2023) on frame rate (fps). For the image-to-video model, SVD replaces text embeddings with the CLIP image embedding (Radford et al., 2021) of the conditioning.

In EDM-framework, the denoiser $\boldsymbol{D}_\theta$ computes the denoised estimate from the U-Net $\boldsymbol{\epsilon}_\theta$:

$$\boldsymbol{D}_\theta(\boldsymbol{x}_t; \sigma, \boldsymbol{c}) = c_{\text{skip}}(\sigma)\boldsymbol{x}_t + c_{\text{out}}(\sigma)\boldsymbol{\epsilon}_\theta\left(c_{\text{in}}(\sigma)\boldsymbol{x}_t; c_{\text{noise}}(\sigma), \boldsymbol{c}\right), \tag{1}$$

where $c_{\text{skip}}$, $c_{\text{out}}$, $c_{\text{in}}$, and $c_{\text{noise}}$ are $\sigma$-dependent preconditioning parameters and $\boldsymbol{c}$ is the condition. In practice, the denoiser $\boldsymbol{D}_\theta$ takes concatenated inputs $[\boldsymbol{x}_t, \boldsymbol{x}_t]$ to return $\boldsymbol{c}$-conditioned estimate and *null*-conditioned estimate $[\hat{\boldsymbol{x}}_{0,c}, \hat{\boldsymbol{x}}_{0,\varnothing}]$, where $\hat{\boldsymbol{x}}_{0,c}$ is then updated using $\omega$-scale classifier-free guidance (CFG) (Ho & Salimans, 2021):

$$\hat{\boldsymbol{x}}_{0,\boldsymbol{c}} \leftarrow \hat{\boldsymbol{x}}_{0,\varnothing} + \omega[\hat{\boldsymbol{x}}_{0,\boldsymbol{c}} - \hat{\boldsymbol{x}}_{0,\varnothing}]. \tag{2}$$

For sampling, SVD employs Euler-step to gradually denoise from Gaussian noise $\boldsymbol{x}_T$ to get $\boldsymbol{x}_0$:

$$\boldsymbol{x}_{t-1,\boldsymbol{c}} = \hat{\boldsymbol{x}}_{0,\boldsymbol{c}} + \frac{\sigma_{t-1}}{\sigma_t}(\boldsymbol{x}_t - \hat{\boldsymbol{x}}_{0,\boldsymbol{c}}), \tag{3}$$

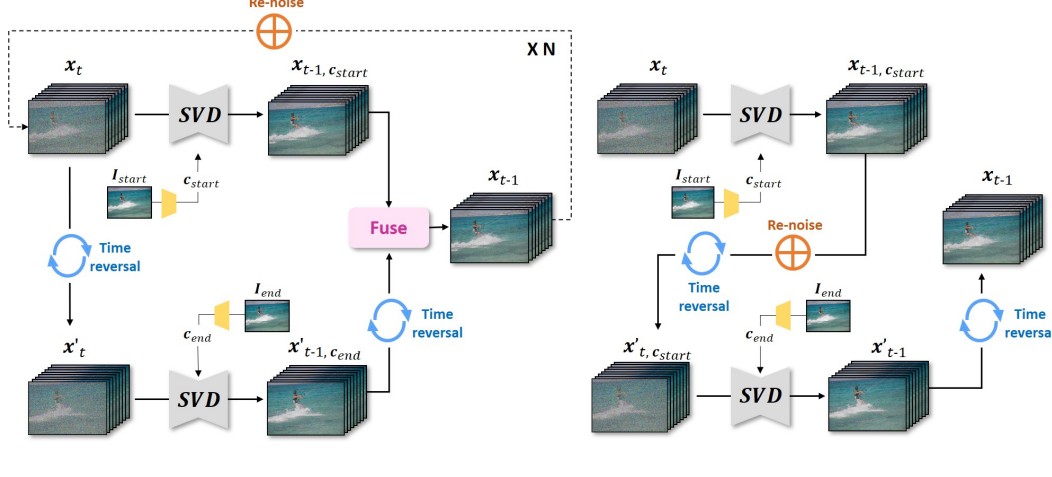

Figure 2: **Comparison of denoising processes.** (a) Time Reversal Fusion method and (b) bidirectional sampling (Ours).

where $\hat{x}_{0,c}$ is the denoised estimate from (2) and $\sigma_t$ is the discretized noise level for each timestep $t \in [0, T]$.

## 3.1 BIDIRECTIONAL SAMPLING

Prior approaches such as TRF (Feng et al., 2024) and Generative Inbetweening (Wang et al., 2024) have employed a fusion strategy that linearly interpolates between samples from the temporally forward path, conditioned on the start frame ($I_{\text{start}}$), and the temporally backward path, conditioned on the end frame ($I_{\text{end}}$):

$$x_{t-1,c_{\text{start}}} = \hat{x}_{0,c_{\text{start}}} + \frac{\sigma_{t-1}}{\sigma_t}(x_t - \hat{x}_{0,c_{\text{start}}}), \tag{4}$$

$$x'_{t-1,c_{\text{end}}} = \hat{x}'_{0,c_{\text{end}}} + \frac{\sigma_{t-1}}{\sigma_t}(x'_t - \hat{x}'_{0,c_{\text{end}}}), \tag{5}$$

$$x_{t-1} = \lambda x_{t-1,c_{\text{start}}} + (1 - \lambda)(x'_{t-1,c_{\text{end}}})', \tag{6}$$

where the $'$ notation indicates that the sample has been flipped along the time dimension, $\lambda$ denotes interpolation ratio, $c_{\text{start}}$ and $c_{\text{end}}$ denotes the encoded latent condition of $I_{\text{start}}$ and $I_{\text{end}}$, respectively. However, as the authors in TRF (Feng et al., 2024) reported, the vanilla implementation of this fusion strategy suffers from random dynamics and unsmooth transitions. This occurs because linearly interpolating between two distinct sample points in the noisy manifold $\mathcal{M}_t$ can cause the deviation from the original manifold, as illustrated in Fig. 3 (a).

In this work, we aim to leverage the image-to-video diffusion model (SVD) for keyframe interpolation tasks, eliminating the multiple noise injections or model fine-tuning. Notably, our key innovation lies in the sequential sampling of the temporally forward path of $x_t$ and the temporally backward path of $x'_t := \text{flip}(x_t)$ by integrating a single re-noising step between them:

$$x_{t-1,c_{\text{start}}} = \hat{x}_{0,c_{\text{start}}} + \frac{\sigma_{t-1}}{\sigma_t}(x_t - \hat{x}_{0,c_{\text{start}}}), \tag{7}$$

$$x_{t,c_{\text{start}}} = x_{t-1,c_{\text{start}}} + \sqrt{\sigma_t^2 - \sigma_{t-1}^2}\,\epsilon, \tag{8}$$

$$x'_{t-1} = \hat{x}_{0,c_{\text{end}}} + \frac{\sigma_{t-1}}{\sigma_t}(x_{t,c_{\text{start}}} - \hat{x}_{0,c_{\text{end}}}), \tag{9}$$

$$x_{t-1} = (x'_{t-1})'. \tag{10}$$

Note that the amount of renoising is determined by the noise difference between $\mathcal{M}_t$ and $\mathcal{M}_{t-1}$, which is crucial to avoid mode collapsing behavior from the forward path sampling. This approach effectively constrains the sampling process for bounded generation between the start frame ($I_{\text{start}}$) and the end frame ($I_{\text{end}}$). As depicted in Fig. 3 (b), our method seamlessly connects the temporally

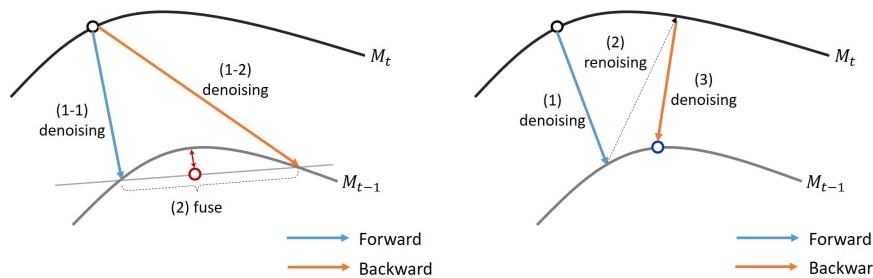

Figure 3: **Comparison of diffusion sampling paths.** (a) Existing methods encounter off-manifold issues due to the averaging of two sample points. (b) In contrast, our bidirectional sampling sequentially connects the temporally forward and backward paths, ensuring that the process remains within the manifold.

forward and backward paths so that the sampling trajectory stays within the SVD manifold, resulting in smooth and coherent transitions throughout the interpolation process.

## 3.2 Additional manifold guidances

We further employ recent advanced manifold guidance techniques to enhance the interpolation performance of the bidirectional sampling. First, we introduce additional frame guidance using DDS (Chung et al., 2023). Then, we replace traditional CFG (Ho & Salimans, 2021) with CFG++ (Chung et al., 2024) to mitigate the off-manifold issue of CFG in the original implementation of SVD (Blattmann et al., 2023a).

**Last frame guidance with DDS.** We introduce additional frame guidance to achieve more accurate bounded generation. Specifically, we employ DDS guidance to align the last frame with $c_{\text{end}}$ in the temporally forward path and with $c_{\text{start}}$ in the temporally backward path. DDS (Chung et al., 2023) synergistically combines the diffusion sampling and Krylov subspace methods (Liesen & Strakos, 2013) such as the conjugate gradient (CG) method, guaranteeing the on-manifold solution of the following optimization problem:

$$\min_{\boldsymbol{x} \in \mathcal{M}} \ell(\boldsymbol{x}) := \|\boldsymbol{y} - \mathcal{A}(\boldsymbol{x})\|^2, \tag{11}$$

where $\mathcal{A}$ is the linear mapping, $\boldsymbol{y}$ is the condition, and $\mathcal{M}$ represents the clean manifold of the diffusion sampling path.

Here, we present a novel insight that DDS, originally designed to address image inverse problems, can also be effectively applied to bounded video generation. Specifically, we define $\mathcal{A}(\boldsymbol{x}) := \boldsymbol{x}_{\text{last}}$ as a last-frame extractor and $\boldsymbol{y}$ as the target condition, which corresponds to $c_{\text{end}}$ for the temporal forward path and $c_{\text{start}}$ for the temporal backward path in bounded video generation. Then we take DDS step from (11) on denoised estimates ($\hat{\boldsymbol{x}}_{0,c_{\text{start}}}$ and $\hat{\boldsymbol{x}}_{0,c_{\text{end}}}$) with target condition ($c_{\text{end}}$ and $c_{\text{start}}$) to get the on-manifold solutions ($\bar{\boldsymbol{x}}_{0,c_{\text{start}}}$ and $\bar{\boldsymbol{x}}_{0,c_{\text{end}}}$) of the following optimization problems:

$$\bar{\boldsymbol{x}}_{0,c_{\text{start}}} = \text{DDS}(\hat{\boldsymbol{x}}_{0,c_{\text{start}}}, c_{\text{end}}) := \underset{\boldsymbol{x} \in \hat{\boldsymbol{x}}_{0,c_{\text{start}}} + \mathcal{K}_l}{\arg\min} \|c_{\text{end}} - \mathcal{A}(\boldsymbol{x})\|^2, \tag{12}$$

$$\bar{\boldsymbol{x}}'_{0,c_{\text{end}}} = \text{DDS}(\hat{\boldsymbol{x}}'_{0,c_{\text{end}}}, c_{\text{start}}) := \underset{\boldsymbol{x} \in \hat{\boldsymbol{x}}'_{0,c_{\text{end}}} + \mathcal{K}_l}{\arg\min} \|c_{\text{start}} - \mathcal{A}(\boldsymbol{x})\|^2, \tag{13}$$

where $\mathcal{K}_l$ is the $l$-th order Krylov subspace, in which Krylov subspace methods seek an approximate solution. By leveraging this DDS framework, we effectively guide the sampling process toward a path conditioned by both the start and end frames, which is particularly effective for keyframe interpolation.

**Better Image-Video alignment with CFG++.** Recent advances of CFG++ (Chung et al., 2024) tackles the inherent off-manifold issue in CFG (Ho & Salimans, 2021). Specifically, CFG++ mitigates this undesirable off-manifold issue using the unconditional score instead of the conditional score in a re-noising process of CFG. By using the unconditional score, CFG++ can overcome the off-manifold phenomena in CFG-generated samples, resulting in better text-image alignment for text-to-image generation tasks.

While SVD replaces text embeddings with CLIP image embeddings, we empirically found that CFG++, initially designed to enhance text alignment in image diffusion models, also significantly improves image-to-video alignment in the video diffusion model. Specifically, after applying CFG++ into SVD sampling framework, the Euler-step of SVD (3) now reads:

$$\boldsymbol{x}_{t-1,\boldsymbol{c}} = \hat{\boldsymbol{x}}_{0,\boldsymbol{c}} + \frac{\sigma_{t-1}}{\sigma_t}(\boldsymbol{x}_t - \hat{\boldsymbol{x}}_{0,\varnothing}), \qquad (14)$$

where the last term in (3) is replaced by $\hat{\boldsymbol{x}}_{0,\varnothing}$. In practice, we apply DDS guidance before CFG++ update, so $\hat{\boldsymbol{x}}_{0,\boldsymbol{c}}$ in (14) should be replaced with $\bar{\boldsymbol{x}}_{0,\boldsymbol{c}}$ as in (12), (13). We experimentally found that incorporating DDS and CFG++ guidance synergistically improves the interpolation performance of bidirectional sampling. The overall sampling method effectively steers the SVD sampling path to perform keyframe interpolation in an on-manifold manner, fully leveraging the generation capabilities of SVD. The detailed algorithm is provided in Algorithm 1. The vanilla bidirectional sampling can be implemented by removing DDS guidance (orange) and replacing the CFG++ update (blue) with a traditional CFG update. The detailed algorithm of the vanilla bidirectional sampling is provided in Appendix A.

---

**Algorithm 1** ViBiDSampler

---

**Require:** $\boldsymbol{x}_T \sim \mathcal{N}(0, \boldsymbol{I}), I_{\text{start}}, I_{\text{end}}, \{\sigma_t\}_{t=1}^T$
1: $\boldsymbol{c}_{\text{start}}, \boldsymbol{c}_{\text{end}} \leftarrow \text{encode}(I_{\text{start}}, I_{\text{end}})$
2: **for** $t = T : 1$ **do**
3: $\quad \hat{\boldsymbol{x}}_{0,\boldsymbol{c}_{\text{start}}}, \hat{\boldsymbol{x}}_{0,\varnothing} \leftarrow \boldsymbol{D}_\theta(\boldsymbol{x}_t; \sigma_t, \boldsymbol{c}_{\text{start}})$ $\qquad\qquad$ ▷ EDM denoised estimate with $\boldsymbol{c}_{\text{start}}$
4: $\quad \bar{\boldsymbol{x}}_{0,\boldsymbol{c}_{\text{start}}} \leftarrow \text{DDS}(\hat{\boldsymbol{x}}_{0,\boldsymbol{c}_{\text{start}}}, \boldsymbol{c}_{\text{end}})$ $\qquad\qquad$ ▷ DDS guidance for end-frame matching
5: $\quad \boldsymbol{x}_{t-1,\boldsymbol{c}_{\text{start}}} \leftarrow \bar{\boldsymbol{x}}_{0,\boldsymbol{c}_{\text{start}}} + \frac{\sigma_{t-1}}{\sigma_t}(\boldsymbol{x}_t - \hat{\boldsymbol{x}}_{0,\varnothing})$ $\qquad\qquad\qquad$ ▷ CFG++ update
6: $\quad \boldsymbol{x}_{t,\boldsymbol{c}_{\text{start}}} \leftarrow \boldsymbol{x}_{t-1,\boldsymbol{c}_{\text{start}}} + \sqrt{\sigma_t^2 - \sigma_{t-1}^2}\epsilon$ $\qquad\qquad\qquad\qquad$ ▷ Re-noise
7: $\quad \boldsymbol{x}'_{t,\boldsymbol{c}_{\text{start}}} \leftarrow \text{flip}(\boldsymbol{x}_{t,\boldsymbol{c}_{\text{start}}})$ $\qquad\qquad\qquad\qquad\qquad$ ▷ Time reverse
8: $\quad \hat{\boldsymbol{x}}_{0,\boldsymbol{c}_{\text{end}}}, \hat{\boldsymbol{x}}'_{0,\varnothing} \leftarrow \boldsymbol{D}_\theta(\boldsymbol{x}'_{t,\boldsymbol{c}_{\text{start}}}; \sigma_t, \boldsymbol{c}_{\text{end}})$ $\qquad\qquad$ ▷ EDM denoised estimate with $\boldsymbol{c}_{\text{end}}$
9: $\quad \bar{\boldsymbol{x}}'_{0,\boldsymbol{c}_{\text{end}}} \leftarrow \text{DDS}(\hat{\boldsymbol{x}}'_{0,\boldsymbol{c}_{\text{end}}}, \boldsymbol{c}_{\text{start}})$ $\qquad\qquad$ ▷ DDS guidance for start-frame matching
10: $\quad \boldsymbol{x}'_{t-1} \leftarrow \bar{\boldsymbol{x}}'_{0,\boldsymbol{c}_{\text{end}}} + \frac{\sigma_{t-1}}{\sigma_t}(\boldsymbol{x}'_{t,\boldsymbol{c}_{\text{start}}} - \hat{\boldsymbol{x}}'_{0,\varnothing})$ $\qquad\qquad\qquad$ ▷ CFG++ update
11: $\quad \boldsymbol{x}_{t-1} \leftarrow \text{flip}(\boldsymbol{x}'_{t-1})$ $\qquad\qquad\qquad\qquad\qquad\qquad$ ▷ Time reverse
12: **end for**
13: **return** $x_0$

---

## 4 EXPERIMENTAL RESULTS

### 4.1 EXPERIMENTAL SETTING

**Dataset.** The high-resolution (1080p) video datasets used for evaluation are sourced from the DAVIS dataset (Pont-Tuset et al., 2017) and the Pexels dataset[1]. For the DAVIS dataset, we pre-processed 100 videos into 100 video-keyframe pairs, with each video consisting of 25 frames. This dataset includes a wide range of large and varied motions, such as surfing, dancing, driving, and airplane flying. For the Pexels dataset, we collected 45 videos, primarily featuring scene motions, natural movements, directional animal movements, and sports actions. We used the first and last frames from each video as keyframes for our evaluation.

**Implementation Details.** For the sampling process, we used the Euler scheduler with 25 timesteps for both forward and backward sampling. The motion bucket ID was fixed at 127, and the decoding frame number was set to 4 due to memory limitations on an NVIDIA RTX 3090 GPU. All other parameters followed the default settings from SVD. Since micro-condition fps is sensitive to the data, we applied a lower fps for cases with large motion and a higher fps for cases with smaller motion. While both DDS and CFG++ generally improve the results, the choice between them depends on the specific use case. All evaluations were performed on a single NVIDIA RTX 3090.

### 4.2 COMPARATIVE STUDIES

We conducted a comparative study with four different keyframe interpolation baselines, including FILM (Reda et al., 2019), a conventional flow-based frame interpolation method, and three frame interpolation methods based on video diffusion models: TRF (Feng et al., 2024), DynamiCrafter (Xing

---

[1] https://www.pexels.com/

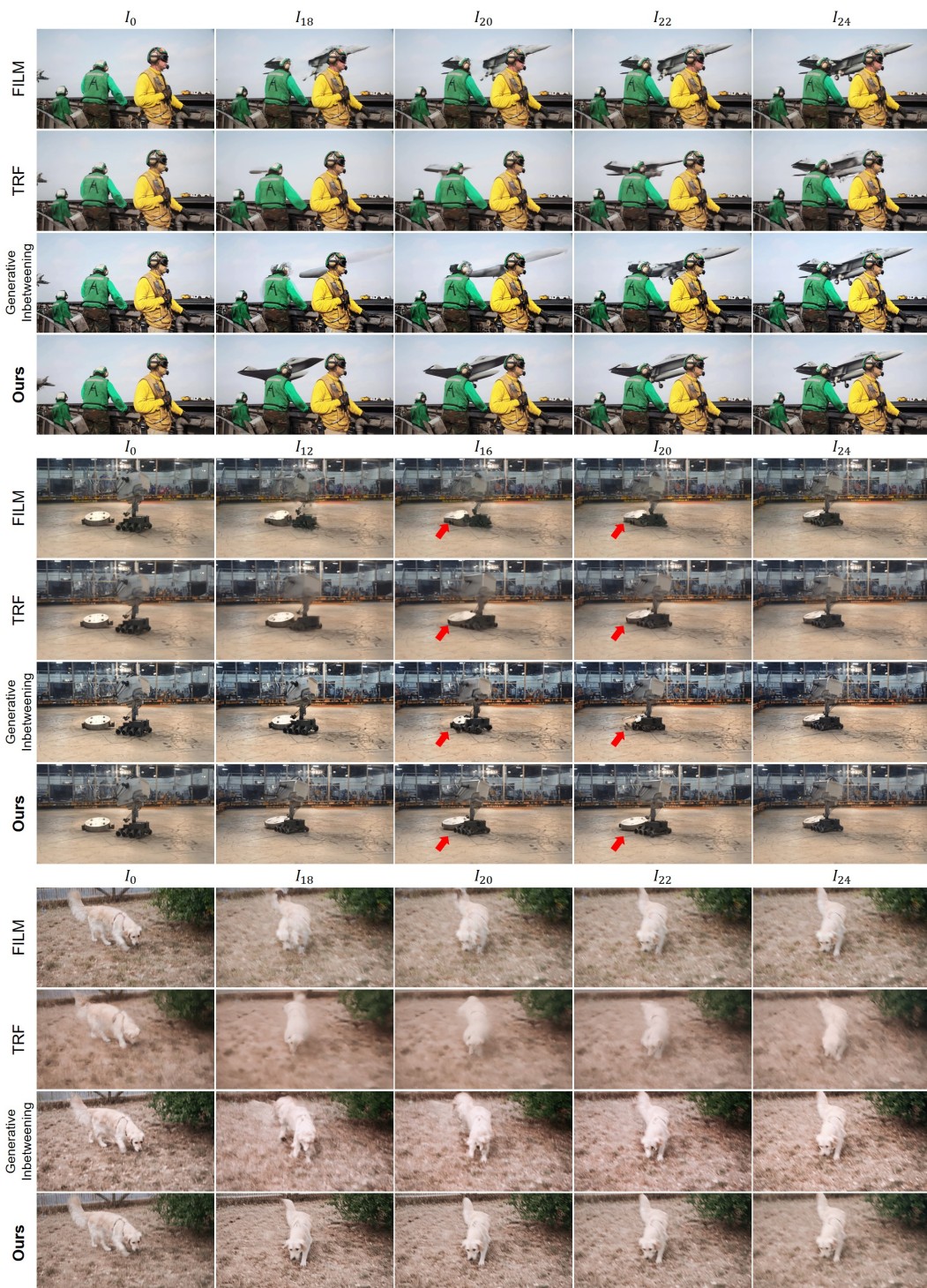

Figure 4: **Qualitative evaluation compared to three baselines: FILM, TRF, and Generative Inbetweening.** The start and end frames ($I_0$, $I_{24}$) are used as keyframes. While FILM encounters difficulties in capturing motion when there is a significant discrepancy between the two keyframes, and TRF and Generative Inbetweening experience a decline in perceptual quality due to the blurring of objects within the image, our method successfully captures motion while maintaining high fidelity in the generated images.

| Method | DAVIS | | | Pexels | | |
|---|---|---|---|---|---|---|
| | LPIPS ↓ | FID ↓ | FVD ↓ | LPIPS ↓ | FID ↓ | FVD ↓ |
| FILM | 0.2697 | 40.241 | 833.80 | **0.0821** | **25.615** | 559.16 |
| TRF [2] | 0.3102 | 60.278 | 622.16 | 0.2222 | 80.618 | 880.97 |
| DynamiCrafter | 0.3274 | 46.854 | 538.36 | 0.1922 | 49.476 | 604.20 |
| Generative Inbetweening | 0.2823 | 36.273 | 490.34 | 0.1523 | 40.470 | 746.26 |
| Ours (Vanilla) | 0.3031 | 52.452 | 543.31 | 0.2074 | 63.241 | 717.37 |
| Ours (Vanilla w/ CFG++) | 0.2571 | 41.960 | 434.41 | 0.1524 | 41.347 | 478.35 |
| **Ours (Full)** | **0.2355** | **35.659** | **399.15** | 0.1366 | 37.341 | **452.34** |

Table 1: **Quantitative evaluation on DAVIS and Pexels datasets.** We compared our method against four different baselines and conducted ablation studies to assess the impact of CFG++ and DDS. *Ours (Vanilla)* refers to the bidirectional sampling method utilizing traditional CFG update without DDS guidance. *Ours (Vanilla w/ CFG++)* refers to the bidirectional sampling method with CFG++ update, also without DDS guidance. **Bold** and underline refer to the best and the second best, respectively.

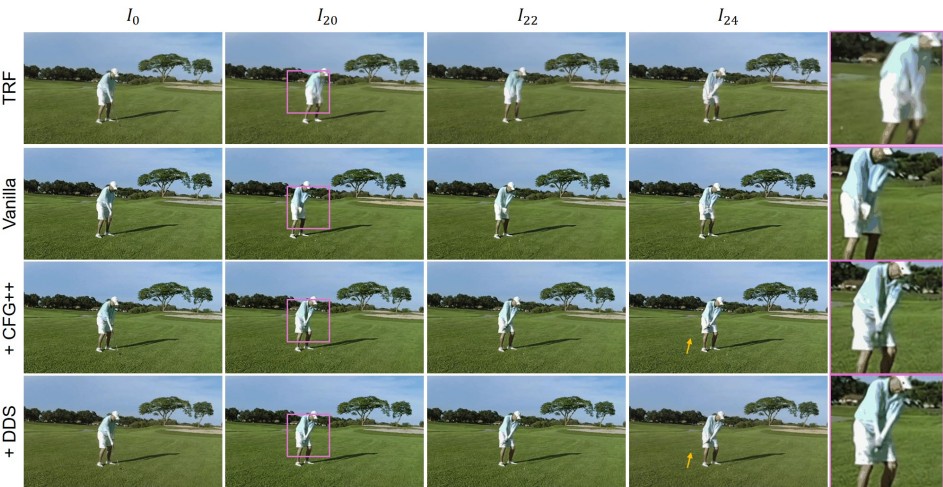

Figure 5: **Ablation study on the effects of CFG++ and DDS.** The inclusion of CFG++ and DDS results in improved perceptual quality in the generated frames.

et al., 2023), and Generative Inbetweening (Wang et al., 2024). We conducted these studies using the official implementations with default values, except for TRF, which has not been open-sourced yet.

**Qualitative evaluation.** As illustrated in Fig. 4, our model clearly outperforms the other methods in terms of motion consistency and identity preservation. Other baselines struggle to accurately predict the motion between the two keyframes when there is a significant difference in content. For example, in Fig. 4, the first frame shows the tip of an airplane, while the last frame reveals the airplane's body. In this case, FILM fails to produce a linear motion path, instead showing the airplane's shape converging toward the middle frame from both end frames, resulting in the airplane's body being disconnected by the 18*th* frame. While TRF and Generative Inbetweening show sequential movement, the airplane's shape becomes distorted. In contrast, our method preserves the airplane's shape while effectively capturing its gradual motion. Furthermore, it can be observed from the second and third cases from Fig. 4 that our method generates temporally coherent results while semantically adhering to the input frames. In TRF, the shapes of the robot and the dog become blurred due to the denoising paths deviating from the manifold during the fusion process, leading to artifacts in the image. While Generative Inbetweening mitigated this off-manifold issue through temporal attention rotation and model fine-tuning, artifacts still persist. In contrast, our method preserves the shapes of both the robot and the dog, generating frames with strong temporal consistency.

**Quantitative evaluation.** For quantitative evaluation, we used LPIPS (Zhang et al., 2018) and FID (Heusel et al., 2017) to assess the quality of the generated frames, and FVD (Unterthiner et al., 2019)

---

[2]Unofficial implementation: `https://github.com/YingHuan-Chen/Time-Reversal`

| Method | NFE | Train | Inference time (s) | Frame # | Resolution |
|---|---|---|---|---|---|
| TRF | 120 | ✗ | 443 | 25 | 1024 × 576 |
| DynamiCrafter | 50 | ✔ | 42 | 16 | 512 × 320 |
| Generative Inbetweening | 300 | ✔ | 1,222 | 25 | 1024 × 576 |
| **Ours** | 50 | ✗ | 195 | 25 | 1024 × 576 |

Table 2: A comprehensive comparison of our method with other diffusion-based approaches.

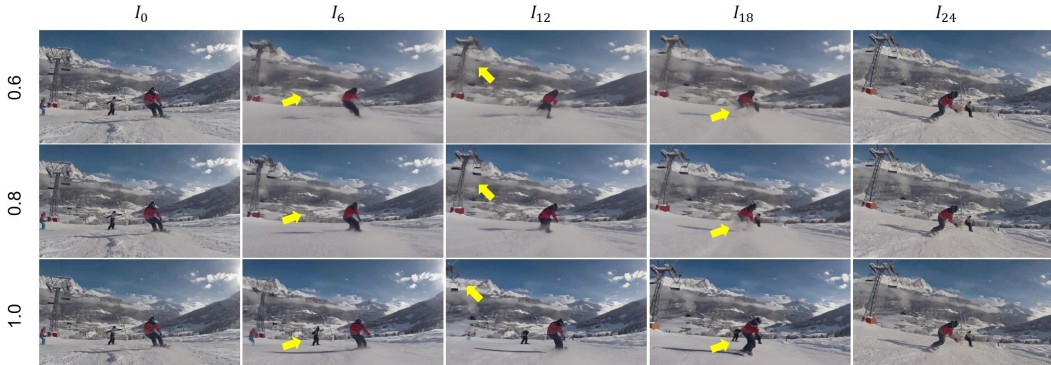

Figure 6: **Effect of CFG++ guidance scale.** The rows, from top to bottom, correspond to the CFG++ scales of 0.6, 0.8, and 1.0.

| Metrics | 0.6 | 0.8 | 1.0 |
|---|---|---|---|
| LPIPS ↓ | 525.36 | 424.03 | **399.15** |
| FID ↓ | 52.5059 | 40.4968 | **35.6594** |
| FVD ↓ | 0.2697 | 0.2394 | **0.2355** |

Table 3: **Quantitative analysis on CFG++ guidance scale** $\omega$**.** Effective results are obtained at the scale of 1.0.

to evaluate the overall quality of the generated videos. As shown in Table 1, our method surpasses the other baselines in terms of fidelity. Moreover, it achieves superior perceptual quality, particularly in scenarios involving dynamic motions (DAVIS), indicating that our approach effectively addresses the issue of deviations from the diffusion manifold, resulting in improved video generation quality. For a more comprehensive comparison, we present quantitative evaluations using PSNR and SSIM metrics in Appendix B.1. Nonetheless, it is important to note that these metrics primarily evaluate pixel-wise accuracy, which does not always align with human perceptual preferences. Within the scope of video generative models that align better with human perceptual preferences, our proposed method outperformed all baseline methods.

## 4.3 COMPUTATIONAL EFFICIENCY

We performed comparative studies on the computational cost of diffusion models, as presented in Table 2. In the training stage, DynamiCrafter undergoes additional training with a large-scale image-to-video model for the frame interpolation task, while Generative Inbetweening also necessitates SVD model fine-tuning, both of which demands significant computational resources. During the inference stage, both TRF and Generative Inbetweening generate videos in $25 \sim 50$ steps for each forward and backward direction, with additional noise injection steps that further increase the number of function evaluations (NFE) and inference time. However, our method does not require additional training or fine-tuning and completes the process in just 25 steps per direction, without requiring multiple re-noising.

## 4.4 ABLATION STUDIES

**Bidirectional sampling and conditional guidance.** The effectiveness of bidirectional sampling can be validated in the vanilla version without any conditional guidance, such as CFG++ or DDS. As demonstrated in Table 1, our vanilla model outperforms TRF across all three metrics, supporting the claim that fusing time-reversed denoising paths leads to off-manifold issues, which our method addresses through bidirectional sampling. In addition, with conditional guidance from CFG++ and

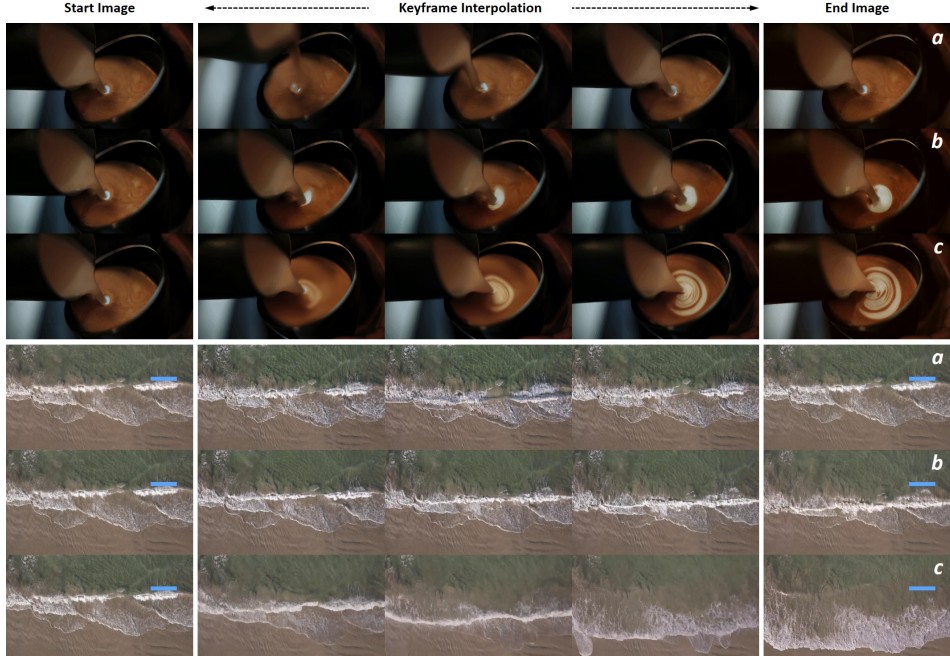

Figure 7: **Application to keyframe interpolation with various boundary conditions.** The end image *a* is identical to the start image. End images *b-c* represent dynamic boundaries sampled from different time points.

DDS, we could achieve even better results and outperform DynamicCrafter and Generative Inbetweening which further train the image-to-video models. This is consistent with Fig. 5, which illustrates that frames generated by TRF exhibit blurry shapes of the golfer and unnecessary camera movement. In contrast, the body shape of the golfer and the golf club are progressively better preserved as additional conditional guidance is incorporated.

**CFG++ guidance scale.** As shown in Fig. 6, at higher CFG++ scales, the semantic information of the input frames is better preserved in the generated intermediate frames, resulting in improved fidelity. For instance, while the small person in the first input frame disappears in the intermediate frames at CFG++ scales of $0.6$ and $0.8$, the person remains visible across all the intermediate frames at a scale of $1.0$. Additionally, as the CFG++ scale decreases, the blurriness of the chairlift in the output frames gradually worsens. This aligns with the findings presented in Table 3. The LPIPS, FID, and FVD values are lowest at a CFG++ scale of $1.0$ and highest at a scale of $0.6$, indicating that CFG++ contributes to improving the perceptual quality of the generated videos.

### 4.5 IDENTICAL AND DYNAMIC BOUNDS

Our method is applicable not only to dynamic bounds, where the start and end frames are different, but also to static bounds, where the start and end frames are identical. As illustrated in Fig. 7, our method successfully generates temporally coherent videos with identical start and end images (*a*). For example, the wave line also consistently fluctuates with the progression of time. Furthermore, as seen in the fifth and sixth rows of Fig. 7, our method effectively generates intermediate frames based on varying end frames (*b* and *c*). Given that the end images of the two rows differ, the resulting intermediate frames are generated accordingly.

### 5 CONCLUSION

We present Video Interpolation using Bidirectional Diffusion Sampler (ViBiDSampler), a novel approach for keyframe interpolation that leverages bidirectional sampling and advanced manifold guidance techniques to address off-manifold issues inherent in time-reversal-fusion-based methods. By performing denoising sequentially in both forward and backward directions and incorporating CFG++ and DDS guidance, ViBiDSampler offers a reliable and efficient framework for generating high-quality, temporally coherent, and vivid video frames without requiring fine-tuning or repeated re-noising steps. Our method achieves state-of-the-art performance in keyframe interpolation, as evidenced by its ability to generate 25-frame video at high resolution in a short processing time.

## 6 ACKNOWLEDGMENT

This work was partly supported by Institute for Information & communications Technology Technology Planning & Evaluation(IITP) grant funded by the Korea government(MSIT)(RS-2019-II190075, Artificial Intelligence Graduate School Support Program(KAIST) and the National Research Foundation of Korea under Grant RS-2024-00336454.

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

## A  ALGORITHM

---

**Algorithm 2** ViBiDSampler (Vanilla)

---

**Require:** $\boldsymbol{x}_T, I_{\text{start}}, I_{\text{end}}, \{\sigma_t\}_{t=1}^T$

1: $\boldsymbol{c}_{\text{start}}, \boldsymbol{c}_{\text{end}} \leftarrow \text{encode}(I_{\text{start}}, I_{\text{end}})$

2: **for** $t = T : 1$ **do**

3:    $\hat{\boldsymbol{x}}_{0,\boldsymbol{c}_{\text{start}}}, \hat{\boldsymbol{x}}_{0,\varnothing} \leftarrow \boldsymbol{D}_\theta(\boldsymbol{x}_t; \sigma_t, \boldsymbol{c}_{\text{start}})$            ▷ EDM denoised estimate with $\boldsymbol{c}_{\text{start}}$

4:    $\boldsymbol{x}_{t-1,\boldsymbol{c}_{\text{start}}} \leftarrow \hat{\boldsymbol{x}}_{0,\boldsymbol{c}_{\text{start}}} + \frac{\sigma_{t-1}}{\sigma_t}(\boldsymbol{x}_t - \hat{\boldsymbol{x}}_{0,\boldsymbol{c}_{\text{start}}})$

5:    $\boldsymbol{x}_{t,\boldsymbol{c}_{\text{start}}} \leftarrow \boldsymbol{x}_{t-1,\boldsymbol{c}_{\text{start}}} + \sqrt{\sigma_t^2 - \sigma_{t-1}^2}\epsilon$                   ▷ Re-noise

6:    $\boldsymbol{x}'_{t,\boldsymbol{c}_{\text{start}}} \leftarrow \text{flip}(\boldsymbol{x}_{t,\boldsymbol{c}_{\text{start}}})$                           ▷ Time reverse

7:    $\hat{\boldsymbol{x}}'_{0,\boldsymbol{c}_{\text{end}}}, \hat{\boldsymbol{x}}'_{0,\varnothing} \leftarrow \boldsymbol{D}_\theta(\boldsymbol{x}'_{t,\boldsymbol{c}_{\text{start}}}; \sigma_t, \boldsymbol{c}_{\text{end}})$      ▷ EDM denoised estimate with $\boldsymbol{c}_{\text{end}}$

8:    $\boldsymbol{x}'_{t-1} \leftarrow \hat{\boldsymbol{x}}'_{0,\boldsymbol{c}_{\text{end}}} + \frac{\sigma_{t-1}}{\sigma_t}(\boldsymbol{x}'_{t,\boldsymbol{c}_{\text{start}}} - \hat{\boldsymbol{x}}'_{0,\boldsymbol{c}_{\text{end}}})$

9:    $\boldsymbol{x}_{t-1} \leftarrow \text{flip}(\boldsymbol{x}'_{t-1})$                               ▷ Time reverse

10: **end for**

11: **return** $x_0$

---

## B  ADDITIONAL RESULTS AND DISCUSSION

### B.1  QUANTITATIVE EVALUATION RESULTS

In addition to the perceptual quality evaluations presented in Table 1, we provide the PSNR and SSIM scores for our method and baseline models to offer a clearer understanding of our work. These metrics were obtained by comparing the generated video frames with their corresponding ground truth frames. As shown in Table 4, FILM achieved the highest PSNR and SSIM scores among the evaluated video generative models, which can be attributed to its training strategy that employs PSNR-oriented loss functions, such as the L1 loss calculated between the ground truth and output frames. However, within the scope of video generative models that align better with human perceptual preferences, our proposed method outperformed all baseline models, recording the highest PSNR and SSIM scores on both the DAVIS and Pexels datasets. These results highlight our model's ability to generate videos that are both visually realistic and exhibit superior fidelity.

| Method | DAVIS | | Pexels | |
|---|---|---|---|---|
| | PSNR ↑ | SSIM ↑ | PSNR ↑ | SSIM ↑ |
| FILM | **22.5521** | **0.5346** | **31.3986** | **0.8418** |
| TRF | 16.4078 | 0.4040 | 19.2670 | 0.5346 |
| Generative Inbetweening | 16.0377 | 0.4255 | 20.8624 | 0.6334 |
| **Ours** | 17.5215 | 0.4387 | 21.4235 | 0.5939 |

Table 4: **Quantitative evaluation on DAVIS and Pexels datasets.** We conducted a comparative analysis of our method with FILM, TRF and Generative Inbetweening in terms of PSNR and SSIM. **Bold** and underline refer to the best and the second best, respectively.

### B.2  ABLATION STUDIES ON ADVANCED GUIDANCES

To emphasize the importance of bidirectional sampling, we conducted additional ablation studies on the guidance techniques, CFG++ and DDS, using the Pexels dataset. These studies involved evaluating the results of the vanilla models and comparing them with the models enhanced by incorporating the guidance techniques.

As shown in Table 5, our vanilla bidirectional sampling consistently outperforms TRF across all metrics. This demonstrates that bidirectional sampling alone effectively addresses off-manifold issues, mitigating the artifacts commonly observed in TRF. Unlike Generative Inbetweening, which requires fine-tuning of the diffusion model, our method operates entirely in a training-free manner.

When comparing models enhanced with guidance techniques, our method demonstrates superior performance compared to both TRF and Generative Inbetweening. Notably, TRF also shows improvements with the addition of guidance techniques, as reflected in the quantitative metrics in Table 5 and the qualitative results available on our anonymous Project page. The results indicate more stable motion and better preservation of bounding frame information. However, its performance remains lower than that of our full method, highlighting the critical role of bidirectional sampling in addressing off-manifold issues and enabling the generation of more accurate and higher-quality videos. In addition, our proposed method exhibits a significantly higher degree of improvement compared to TRF. Since manifold guidance methods, such as CFG++, were originally designed for on-manifold sampling, our method can be synergistically enhanced with these guidance techniques for further improvement. In contrast, TRF faces challenges in achieving this synergistic improvement due to the generation of off-manifold samples, highlighting the differences in performance and effectiveness between the two approaches. In contrast, the integration of guidance techniques into Generative Inbetweening does not yield substantial improvements. This limited effect may be due to the interference between its fine-tuning process and rotated temporal attention, which could hinder the effectiveness of the guidance techniques.

We strongly encourage readers to visit our anonymous Website for a deeper understanding of the distinct roles of bidirectional sampling and guidance methods.

| Method | Vanilla | | | CFG++ & DDS | | |
|---|---|---|---|---|---|---|
| | LPIPS $\downarrow$ | FID $\downarrow$ | FVD $\downarrow$ | LPIPS $\downarrow$ | FID $\downarrow$ | FVD $\downarrow$ |
| TRF | 0.2222 | 80.618 | 880.97 | 0.2010 | 52.738 | 778.69 |
| Generative Inbetweening | 0.1523 | 40.470 | 746.26 | 0.1662 | 42.487 | 747.95 |
| **Ours** | 0.2074 | 63.241 | 717.37 | **0.1366** | **37.341** | **452.34** |

Table 5: **Ablation study on CFG++ and DDS**. **Bold** and underline refer to the best and the second best, respectively.

### B.3 Discussions on the roles of CFG++ and DDS

**CFG++.** CFG++ is a guidance technique designed to improve alignment between images and text conditions. By applying CFG++ guidance to our model, the generated videos achieve better alignment with the image conditions and maintain semantic consistency with the given frames, $I_{start}$, and $I_{end}$. This enhancement positively impacts the perceptual quality of the videos.

**DDS.** SVD, an image-to-video diffusion model, generates 25-frame videos based on an initial frame condition, without utilizing a final frame condition. To address the absence of a last frame condition within the video interpolation framework, we incorporated DDS guidance. This approach ensures alignment of the last frame with $c_{end}$ in the temporally forward path and $c_{start}$ in the temporally backward path, thereby enhancing the temporal consistency of the generated videos.

### B.4 Ablation study on SVD version.

Since we employed SVD-XT, which is the fine-tuned version of the original SVD, we checked the performance of our method with SVD. As shown in the Table 6, we see the performance increment in better video generation models.

| Method (Pexels) | FVD $\downarrow$ | LPIPS $\downarrow$ |
|---|---|---|
| SVD | 608.07 | 0.1577 |
| SVD-XT (ours) | **452.34** | **0.1366** |

Table 6: **Ablation study on SVD version. Bold** refer to the best.

### B.5 Future work and discussions

Our proposed method demonstrates exceptional performance in generating intermediate frames from two bounding frames. However, as the model is based on Stable Video Diffusion, which employs

image embeddings as conditions for cross-attention instead of textual prompts, we have not considered the incorporation of text conditioning. Nonetheless, when bidirectional sampling is applied to other image-to-video (I2V) diffusion models, such as DynamiCrafter, we believe that it becomes feasible to guide actions based on textual conditions. Extending our method to support text-based control presents a promising avenue for future research.

## B.6 ADDITIONAL EXPERIMENTAL RESULTS

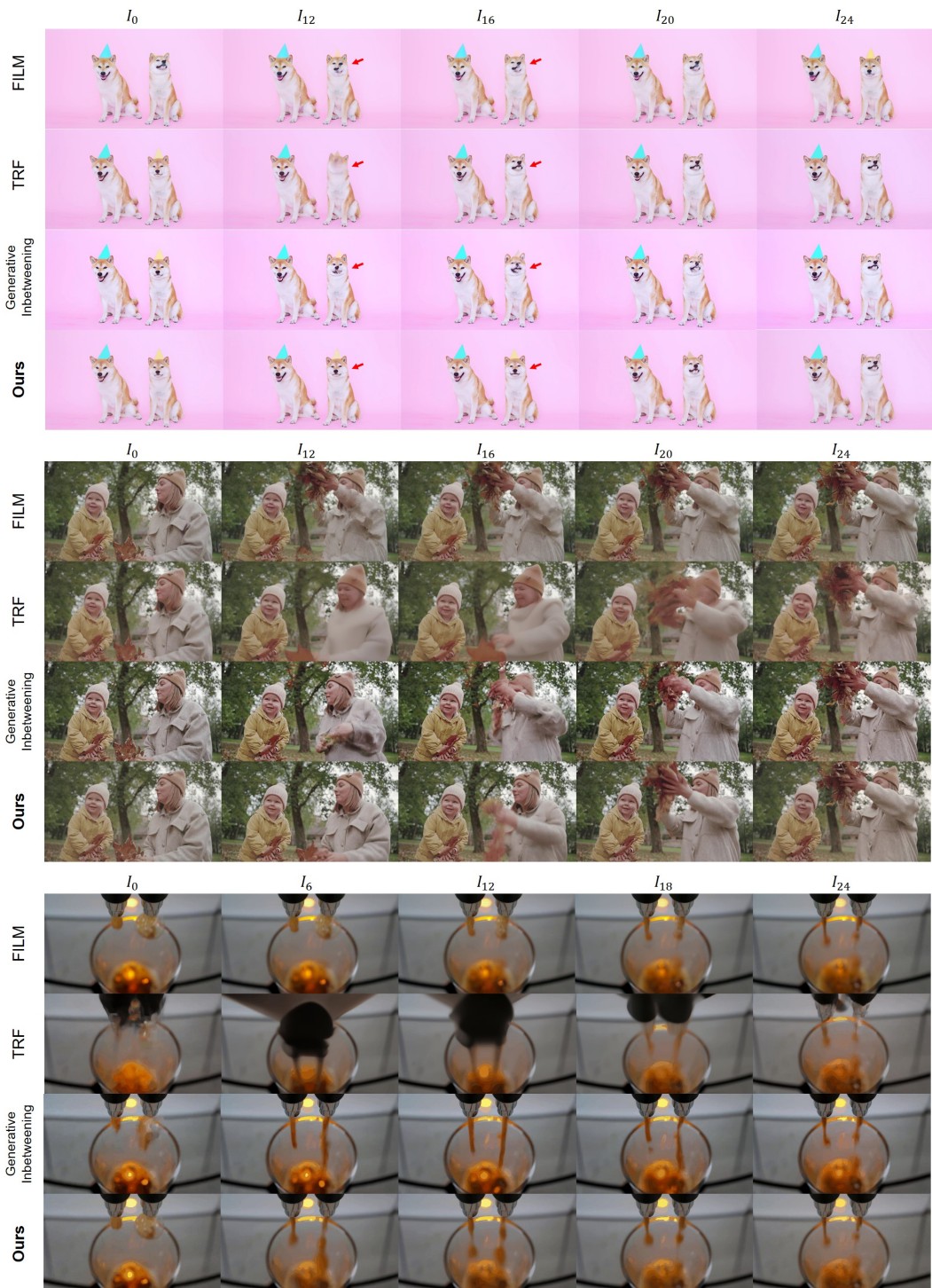

Figure 8: Additional comparison with baseline methods.

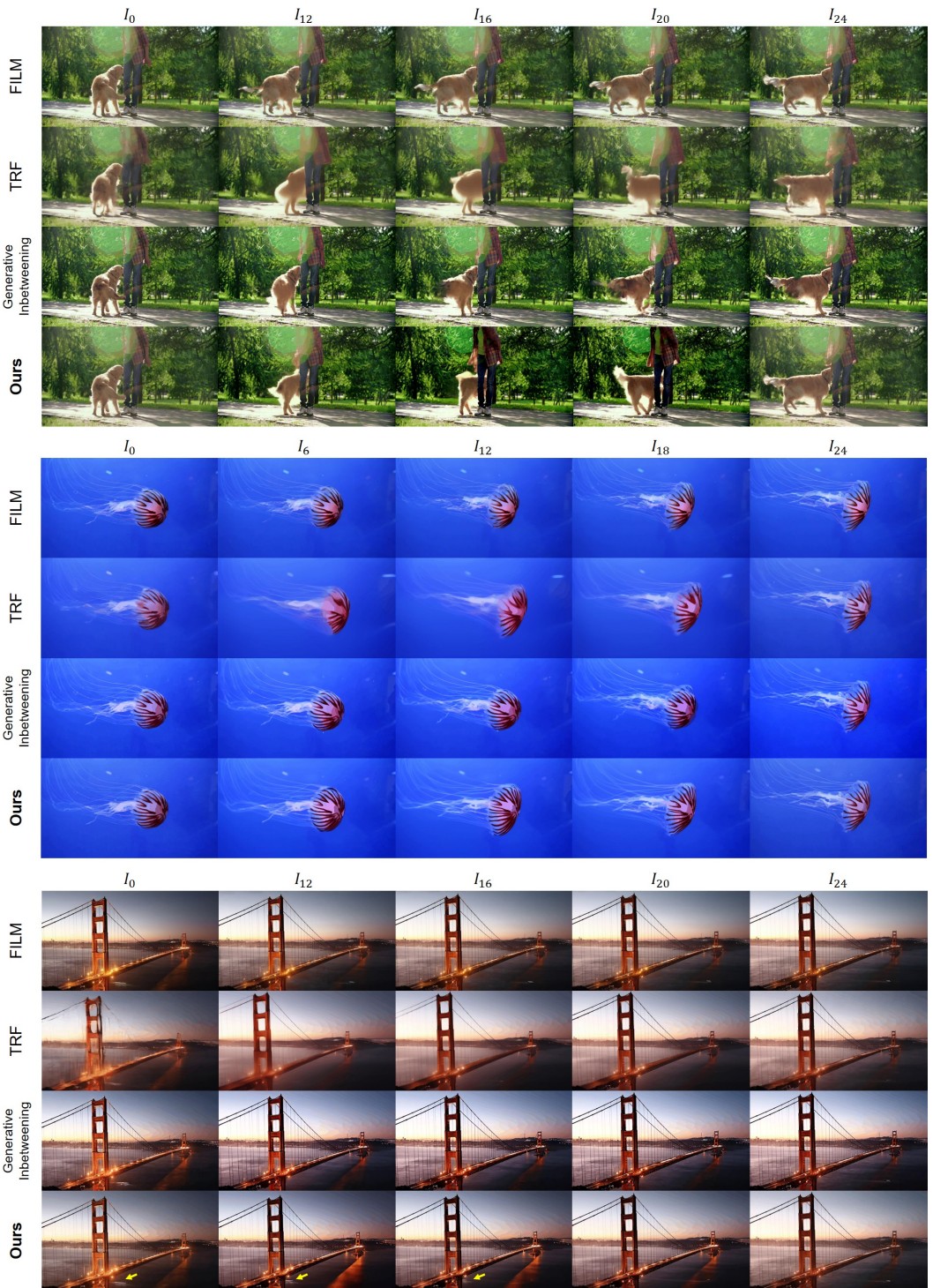

Figure 9: Additional comparison with baseline methods.

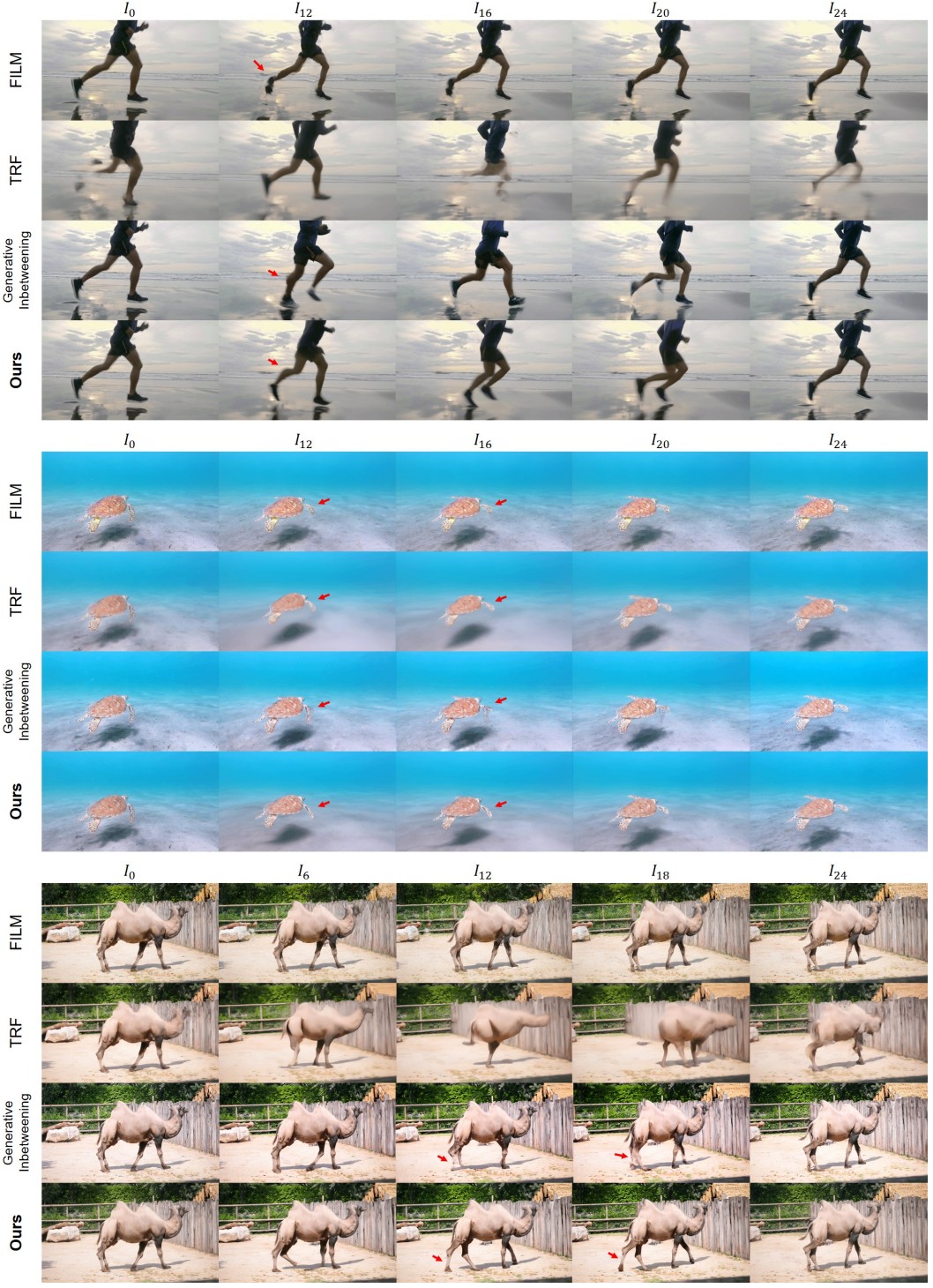

Figure 10: Additional comparison with baseline methods.

