# OpenReview forum: "ViBiDSampler: Enhancing Video Interpolation Using Bidirectional Diffusion Sampler"
_ICLR.cc/2025/Conference — ICLR 2025 Poster_

### Official Review · Reviewer_hsxN · 2024-10-29

**Soundness:** 3
**Presentation:** 3
**Contribution:** 2
**Rating:** 6
**Confidence:** 3

**Summary:**

This paper proposes ViBiDSampler, a training-free method utilizing pre-trained image-to-video diffusion models for keyframe interpolation. They aim to tackle the off-manifold problem of previous studies, which leads to artifacts or costly processes. They propose bidirectional sampling, with a set of denoising-renoising-denoising with alternating conditioning on start and end frames. This way, they manage to keep the generated results stay on the original manifold. The additionally integrate advanced guidance methods like CFG++ and DDS to reach state-of-the-art performance.

**Strengths:**

Strengths
- Overall, the paper is easy to follow, with coherent and clear problem definition and solution.
- Simple methodology. The proposed method, Bidirectional sampling seems to be easy to use.
- Efficiency. The proposed method does not require additional training and is fast in inference.

**Weaknesses:**

1. Notations.
- The notations are confusing. For instance, $x_t$ seems to be used as a variable and a function.
- The definitions could be further clarified.
- Lines 247 - 256 was kind of confusing at first, with many similar looking notations without predefinitions, and mixed use of notations as functions and variables.


2. Limited contribution.

Although the proposed method is intuitive, simple, and quantitatively effective, some uncertainties remain.
- the 'Vanilla' setting at Table 1 and Figure 5, seems to show the effectiveness of bidirectional sampling solely, without guidance such as CFG++ and DDS. However, I am not fully convinced with its effectiveness, as Generative Inbetweening seems to outperform the vanilla version in Table 1. Also, in the qualitative results of Figure 5, the results of bidirectional sampling (vanilla) itself does not seem to be better than that of TRF.
- The concern above makes me wonder, if the superior performance of the entire framework proposed comes from the good guidance methods, CFG++ and DDS.
- In addition, CFG++ and DDS are manifold guidance methods proposed in other studies, and seems like this work simply took advantage of existing methods. I think it is hard to say that simply adoption of these guidance methods is a contribution.

In short, I am not fully convinced if the proposed bidirectional sampling itself is a powerful method compared to existing works, and suspect the superior performance could have come from the advanced manifold guidance methods, which, is hard to say to be a contribution.

**Questions:**

1. To my understanding, CFG++ and DDS could possibly be applied to existing methods, such as Generative Inbetweening. If so, could the authors provide experiments on this, to ablate and show the importance of bidirectional sampling?

2. In bidirectional sampling, in some aspects, I think it is possible for this to work, but in some aspects, I don't. For example, taking the example of Fig.3(b), there are three steps, (1) denoising (w/ $c_{start}$), (2) renoising, (3) denoising (w/ $c_{end}$). I am curious how the generation process goes through these steps. Maybe a visualization would be helpful, showing how the start / end conditions are really affecting in those steps. I am specifically curious of this process in that the (2) renoising step could possibly remove the information of $c_{start}$ injected from (1) denoising, and the conditioning could have possibly be injected only at (3), making the steps (1) and (2) redundant. A further analysis / clarification on this part, that the proposed method surely takes both start / end frame conditioning.

---

> ### Author Response · Authors · 2024-11-20
> **Reply to reviewer hsxN**
>
> **W1. Notations: Lines 247 - 256 (Section 3.2, last frame guidance with DDS) was kind of confusing at first, with many similar looking notations without predefinitions, and mixed use of notations as functions and variables.**
>
> **A.** In response to the reviewer’s request, we modified the mixed use of notations as functions and variables. We further revised confusing definitions. Please refer to the revised Section 3.
>
> **W2, Q1. I am not fully convinced if the proposed bidirectional sampling itself is a powerful method compared to existing works, and suspect the superior performance could have come from the advanced manifold guidance methods, which, is hard to say to be a contribution. Could the authors provide experiments on this, to ablate and show the importance of bidirectional sampling?**
>
> **A.** Thank you for your constructive suggestions that highlight our contributions. First, we would like to inform the reviewer that we have already demonstrated the superior performance of our bidirectional sampling method, referred to as “Ours (Vanilla),” compared to time reversal fusion (TRF) across all experiments and metrics. Unlike Generative Inbetweening, which requires fine-tuning of the diffusion model, our method operates entirely in a training-free manner.
>
> Moreover, we have conducted additional ablation studies and included further visualizations to demonstrate the effect of each building block. Please refer to the additional project page (https://vibid.github.io/supple/) and the Appendix B.2.
>
> As illustrated in the first section of our anonymous webpage, the vanilla TRF model exhibits unnatural motion in videos featuring a coffee machine and a child kicking a ball. In contrast, our vanilla model produces more natural motion and visually vivid frames compared to the TRF vanilla model. We also agree that when advanced guidance techniques, such as CFG++ and DDS, are applied to TRF and our method, video quality of both methods improves. Nonetheless, the videos generated by the TRF model with these guidance techniques remain inferior to our model. These qualitative observations are consistent with the quantitative analysis presented in the Table 5.
>
> Specifically, as shown in the second section of our anonymous project page, the goldfish positioned on the far left in the vanilla TRF model disappears in the intermediate frames. In contrast, the fish exhibits natural movement in the videos generated by our vanilla model. Additionally, while the camel's shape appears blurred and its movement lacks smoothness in videos produced by the vanilla TRF model, our vanilla model demonstrates improved clarity in the camel's form. Furthermore, when advanced guidance techniques are applied, the videos display enhanced quality in the camel's leg shapes and movements.
>
>
> | Vanilla | LPIPS ↓ | FID ↓ | FVD ↓ | | CFG++ & DDS |  LPIPS ↓ | FID ↓ | FVD ↓ |
> |------------------ |:------------------:|:-----------------:|:-----------------:|:-----------------:|:------------------|:-----------------:|:-----------------:|:-----------------:|
> | TRF | 0.2222 | 80.618 | 880.97 | | TRF     | 0.2010 | 52.738 | 778.69 |
> | Generative Inbetweening | 0.1523 | 40.470 | 746.26 | | Generative Inbetweening    | 0.1662 | 42.487 | 747.95 |
> | **Ours** | 0.2074| 63.241 | 717.37 |  |**Ours**  | **0.1366** | **37.341** | **452.34** |
>
>
> **Q2. There are three steps, (1) denoising (w/ c_start), (2) renoising, (3) denoising (w/ c_end). I am specifically curious of this process in that the (2) renoising step could possibly remove the information from (1) denoising, and the conditioning could have possibly be injected only at (3), making the steps (1) and (2) redundant. A further analysis / clarification on this part, that the proposed method surely takes both start / end frame conditioning.**
>
> **A.** The renoising step is not redundant but rather a critical component of bidirectional sampling. During the renoising step, a small amount of additional stochasticity is introduced (Eq. 8), which plays a key role in preventing mode collapse problem by facilitating the effective integration of information from both the forward and backward paths.
>
> For easy understanding, we additionally conducted a simple comparison using only the (3) denoising step (w/ $c_{end}$) as suggested by Reviewer. As shown in the Table, our method outperforms the approach using only the (3) denoising step (w/ $c_{end}$). This ablation study confirms that our slight renoising step retains the information from the (1) denoising step.
>
> | Steps | PSNR ↑ | SSIM ↑ | LPIPS↓ | FVD ↓|
> |------------------ |:------------------:|:-----------------:|:-----------------:|:-----------------:|
> | Ours w/o (1),(2)| 18.5487 | 0.4799 | 0.2183 | 652.32 |
> | **Ours** | **21.4235**| **0.5939** | **0.1366** | **452.34** |

---

> > ### Comment · Reviewer_hsxN · 2024-11-22
> >
> > Thank you for the reply. I have read the full response and have seen the additional project page.
> > I have a few more questions remaining.
> >
> > 1. Experimentally, Generative Inbetweening still seems to perform better compared to the proposed method, without any advanced guidance techniques such as CFG++ and DDS. However, when adopting the advanced guidance techniques, the proposed method seems to take good advantage of it better than Generative Inbetweening. In the case of Generative Inbetweening, the performance rather deteriorates with such techniques. Why would Generative Inbetweening fail in adopting such techniques? Better applicability of existing techniques on a method is a strong advantage, but I believe there should also be an explanation on why some prior studies fail to do so for comparison.
> >
> > 2. In Q2, the authors have shown experimental results with a short explanation on the necessity of steps (1) and (2). Although the experimental results do confirm its effectiveness, the explanation provided by the authors still do not fully explain the explicit roles of each step. Mode collapse problem may be avoided, but I think a solution to mode collapse does not necessarily lead to better reconstruction / generation quality, which, the authors provided as a support for their argument. Thus I think the experimental result provided by the authors and the explanation given is not perfectly aligned. Could the authors provide a further clarified explanation / analysis on this? As it is (to my understanding) the most important / novel contribution of the paper, I believe a deeper study is essential.

---

> > > ### Author Response · Authors · 2024-11-22
> > >
> > > **Response for Q1.**
> > >
> > > Thanks for asking this important question that highlight advantages of our method. Unlike TRF and our method, Generative Inbetweening introduces an additional condition—rotated temporal self-attention—which requires fine-tuning. This fine-tuning increases the diffusion model's reliance on the additional condition. As CFG++ is designed to improve alignment with image conditions rather than rotated temporal self-attention, its effectiveness in this context may be limited. This is evident from the performance improvement observed in TRF, which can be regarded as a non-fine-tuned version of Generative Inbetweening that does not use reversed temporal self-attention.
> > >
> > > **Response for Q2.**
> > >
> > > Thank you for raising this important question. We kindly remind the reviewer that pixel-wise addition of independent Gaussian noise corresponds to the convolution of the probability density function with a blur kernel, as shown in [1]. This implies that the resulting noisy distribution is essentially a blurred version of the original distribution, effectively eliminating potential artifacts in the forward denoising direction. Accordingly, successive applications of backward directional denoising can begin from a distribution unaffected by these artifacts. Without this "renoising" step, artifacts generated in the forward direction are difficult to remove, leading to what we refer to as "mode collapsing" behavior. This observation is supported by empirical evidence from SDEdit [2], which demonstrates that while small additive noise levels preserve the faithfulness to the original image, the introduced stochasticity provides an opportunity to edit the image according to the specified condition.
> > >
> > > To clarify, the specific roles of each step are as follows:
> > >
> > > (1) Denoising (with $c_{start}$): Incorporates information from  $c_{start}$ into the denoised sample.
> > >
> > > (2) Renoising: Adjusts the sample back to the original manifold, keeping it close to the denoised sample while introducing stochasticity to remove potential artifact from the forward directional denoising. Accordingly, this stochasticity facilitates the effective integration of the two denoising paths.
> > >
> > > (3) Denoising (with  $c_{end}$): Incorporates information from  $c_{end}$ into the denoised sample.
> > >
> > > **References**
> > >
> > > [1] Chung, H., et al. “Improving diffusion models for inverse problems using manifold constraints.” NeurIPS 2022.
> > >
> > > [2] Meng, C., et al. “SDEdit: Guided image synthesis and editing with stochastic differential equations.” ICLR 2022.

---

> > > > ### Comment · Reviewer_hsxN · 2024-11-25
> > > >
> > > > Thank you for the detailed response.
> > > > I am satisfied with the rebuttal and have raised the score.

---

> > > > > ### Author Response · Authors · 2024-11-25
> > > > >
> > > > > Thank you for the constructive comments and raising the score!

---

### Official Review · Reviewer_hjqi · 2024-10-31

**Soundness:** 3
**Presentation:** 3
**Contribution:** 2
**Rating:** 6
**Confidence:** 4

**Summary:**

This paper proposes a inference strategy to achieve frame interpolation w/o training. Its key idea is to use the pre-trained stable video diffusion model as a prior and complete the frame interpolation given the first and the last frame as a condition. The main improvement seems from the proposed bidirectional sampling combining with SOTA sampling approaches, i.e., DDS and CFG++.

**Strengths:**

+The paper is easy to follow .
+The visualization results seem impressive comparing with other baselines provided in the paper.

**Weaknesses:**

The contribution of this paper seems marginal. It seems the main contribution comes from eq. (9). However, the improvement of such modification seems not that obvious comparing with using SOTA sampling approaches including DDS and CFG++ from Table. 1, i.e., only "Ours (full)" shows obvious improvement comparing with other baselines but if my understanding is correct, other baselines seem not using DDS and CFG++.

**Questions:**

My main concerns are mainly as follows:

1. The contribution of this paper seems marginal. It seems the main contribution comes from eq. (9). However, the improvement of such modification seems not that obvious comparing with using SOTA sampling approaches including DDS and CFG++ from Table. 1, i.e., only "Ours (full)" shows obvious improvement comparing with other baselines but if my understanding is correct, other baselines seem not using DDS and CFG++. The author should further verify the improvement of each technology used in the paper, i.e., bidirectional sampling, DDS and CFG++, e.g., providing more visualization as well as adding DDS and CFG++ to other baselines and show the improvement.

2. The practical use the proposed approach seems not convincing. The proposed approach still requires 192s to generate 25 frames, which is at least two times longer than directly inferencing a video generation model with similar computational cost. Moreover, I also wonder how to keep the consistency between frames when the required length of the output video is beyond 25.

---

> ### Author Response · Authors · 2024-11-20
> **Reply to reviewer hjqi**
>
> **W1, Q1. The improvement of contribution from bidirectional sampling seems not obvious with using SOTA sampling approaches including DDS and CFG++ from Table. 1. The author should further verify the improvement of each building blocks used in the paper, i.e., bidirectional sampling, DDS and CFG++, e.g., providing more visualization as well as adding DDS and CFG++ to other baselines and show the improvement.**
>
> **A.** Thank you for the constructive suggestion that highlights our contributions. Firstly, reviewer is kindly reminded that, as shown in the Table 1, we have already demonstrated the superior performance of our bidirectional sampling, referred to as “Ours (Vanilla)”, compared to the time reversal fusion (TRF) across all experiments and metrics. Notably, Generative Inbetweening conducts additional fine-tuning on diffusion model to improve the performance, whereas our method is conducted efficiently in training-free manner.
>
> Moreover, we have conducted additional ablation studies and included further visualizations to demonstrate the effect of each building block. Please refer to the additional project page (https://vibid.github.io/supple/) and the Appendix B.2.
>
> As illustrated in the first section of our anonymous webpage, the vanilla TRF model exhibits unnatural motion in videos featuring a coffee machine and a child kicking a ball. In contrast, our vanilla model produces more natural motion and visually vivid frames compared to the TRF vanilla model. We also agree that when advanced guidance techniques, such as CFG++ and DDS, are applied to TRF and our method, video quality of both methods improves. Nonetheless, the videos generated by the TRF model with these guidance techniques remain inferior to our model. These qualitative observations are consistent with the quantitative analysis presented in the Table 5.
>
> Specifically, as shown in the second section of our anonymous project page, the goldfish positioned on the far left in the vanilla TRF model disappears in the intermediate frames. In contrast, the fish exhibits natural movement in the videos generated by our vanilla model. Additionally, while the camel's shape appears blurred and its movement lacks smoothness in videos produced by the vanilla TRF model, our vanilla model demonstrates improved clarity in the camel's form. Furthermore, when advanced guidance techniques are applied, the videos display enhanced quality in the camel's leg shapes and movements.
>
>
>
> | Vanilla | LPIPS ↓ | FID ↓ | FVD ↓ | | CFG++ & DDS |  LPIPS ↓ | FID ↓ | FVD ↓ |
> |------------------ |:------------------:|:-----------------:|:-----------------:|:-----------------:|:------------------|:-----------------:|:-----------------:|:-----------------:|
> | TRF | 0.2222 | 80.618 | 880.97 | | TRF     | 0.2010 | 52.738 | 778.69 |
> | Generative Inbetweening | 0.1523 | 40.470 | 746.26 | | Generative Inbetweening    | 0.1662 | 42.487 | 747.95 |
> | **Ours** | 0.2074| 63.241 | 717.37 |  |**Ours**  | **0.1366** | **37.341** | **452.34** |
>
> **Q2-1. The practical use the proposed approach seems not convincing. The proposed approach still requires 192s to generate 25 frames, which is at least two times longer than directly inferencing a video generation model with similar computational cost.**
>
> **A.** The reviewer is kindly reminded that the video interpolation problem is a bit different from video generation. Video generation is more focused on “fake” video generation based on conditions, whereas video interpolation technique could be applied to interpolate between “real” video frames. Therefore, reviewer is kindly reminded that in this real-world scenario applying video generation model instead of interpolation is not a viable option.
>
> Moreover, we emphasize that our method's inference time, which is only 2× that of direct video generation, is significantly more efficient compared to other VDM-based methods (e.g., TRF with 4.5× and Generative Inbetweening with 12.5× inference time of direct video generation).
>
>
> **Q2-2. How to keep the consistency between frames when the required length of the output video is beyond 25?**
>
> **A.** Thanks for the constructive comments that highlight our advantages. Note that recursive sampling can be used to ensure consistency when the desired output video length exceeds the trained configuration. Specifically, once 25 frames are generated at the first stage, we can further improve the temporal resolution between frames by recursively applying our sampler.
>
> If reviewer’s intension is to extend beyond the two end frames, we would like to kindly remind the reviewer that the suggested problem is not video interpolation; rather, it is a video extrapolation problem which is beyond the scope of this paper and other existing video interpolation methods.

---

> > ### Comment · Reviewer_hjqi · 2024-11-23
> >
> > The author partially addressed my concerns, so I increased my score.
> > Additionally, I would explain my previous comments a little bit:
> >
> > 1. I understood that Ours (Vanilla) shows better performance than TRF, i.e., LPIPS 0.2222 --> 0.2074. However, adding CFG++ and DDS leads to more obvious improvement: 0.2010 --> 0.1366. The author is suggested to provide more explanation on why the proposed approach aligns with CFG++ and DDS better in the paper.
> >
> > 2. What I mean is actually that finetuning a generative model into an interpolation model is easy, i.e., adding an additional condition concatenated with the input is enough, which leads to almost no additional parameter and inference cost. Thus, the proposed approach from my view, still struggles in real-world applications due to the slow inference speed. Such a problem can be more obvious when applying to larger models, i.e., 7B. I understand this is an inherent problem for such a kind of zero-shot paradigm, but the significance of such a line of works still needs further verification.
> >
> > 3. I am not sure if applying the proposed method repeatedly on a given frame sequence too many times can still maintain good quality on the interpolated frames. The performance degradation on conditional generation, i.e., generating future frames conditioned on part of previous frames is a common issue. Maybe such a phenomenon is not that obvious for interpolation since generating middle frames based on former and later frames can be slightly easier.

---

> > > ### Author Response · Authors · 2024-11-24
> > >
> > > Thank you for raising the score. We are glad to hear that our revisions and feedback have successfully addressed your initial concerns. Regarding the additional comments, we have provided our responses below.
> > >
> > > **Response to Q1.**
> > >
> > > Thank you for raising this important question, which provides an opportunity to emphasize the advantages of our method. Unlike TRF, our approach ensures the on-manifold generation of intermediate frames. Since manifold guidance methods, such as CFG++, were originally designed for on-manifold sampling, we believe that our method can be synergistically enhanced with these guidance techniques for further improvement.
> > >
> > > In contrast, TRF faces challenges in achieving this synergistic improvement due to the generation of off-manifold samples, highlighting the differences in performance and effectiveness between the two approaches.
> > >
> > > **Response to Q2.**
> > >
> > > We agree that fine-tuning a generative model specifically for video frame interpolation can be effective in reducing inference costs. That said, one significant drawback of fine-tuning is the requirement for additional training, which demands substantial amounts of data and computational resources that may not be accessible to many researchers. In this context, an inference-time zero-shot interpolation approach offers a practical alternative, as it eliminates the need for additional overhead. Importantly, the proposed zero-shot method still achieves state-of-the-art (SOTA) performance despite the absence of fine-tuning.
> > >
> > > We would like to emphasize to the reviewer that our intention is not to compete with fine-tuning-based approaches. Instead, our goal is to provide a complementary and competitive solution for users when fine-tuning is not a feasible option.
> > >
> > >
> > > **Response to Q3.**
> > >
> > > Thank you for highlighting this critical issue, which highlights the advantages of bidirectional sampling. As noted by the reviewer, video generation fundamentally differs from video interpolation. In video generation, as the number of output frames increases, generating future frames conditioned only on the previous frame becomes progressively more difficult due to the growing distance from the initial condition.
> > >
> > > In contrast, video interpolation benefits from the presence of an end frame. Specifically, as an intermediate frame gets farther from the initial frame, it becomes closer to the end frame, simplifying its generation when conditioned on the end frame. The proposed bidirectional sampling approach leverages this by synergistically sampling in both directions to improve the interpolation process.
> > >
> > > Regarding the repeated recursive application of the method to further increase temporal resolution, we agree with the reviewer that potential performance degradation might occur due to the propagation of interpolation errors. We acknowledge that this is a universal challenge for most generative models and believe that addressing this issue will require further investigation in future research.

---

### Official Review · Reviewer_wFk9 · 2024-11-04

**Soundness:** 3
**Presentation:** 3
**Contribution:** 3
**Rating:** 8
**Confidence:** 4

**Summary:**

This paper presented a bidirectional sampling strategy, sequential sampling along both forward and backward paths, for video interpolation. Additional manifold guidance such as DDS and CFG++ are further introduced to enhance the interpolation results. Experiments are conducted to evaluate the proposed ViBiDSampler.

**Strengths:**

+ A bidirectional sampling strategy, sequential sampling along both forward and backward paths, for video interpolation.
+ Additional manifold guidance such as DDS and CFG++ are further introduced to enhance the interpolation results.
+ Training-free, convenient for use.
+ Experiments are conducted to evaluate the proposed ViBiDSampler.

**Weaknesses:**

- The method is training-free. In general, training-based method can be more effective.
- More ablation study is suggested to assess the effect of pre-trained video generation models. Will better video generation models result in better performance?
- Discussion on the limitation and future work.

**Questions:**

- The method is training-free. In general, training-based method can be more effective.
- More ablation study is suggested to assess the effect of pre-trained video generation models. Will better video generation models result in better performance?
- Discussion on the limitation and future work.

---

> ### Author Response · Authors · 2024-11-20
> **Reply to reviewer wFk9**
>
> **W1, Q1. The method is training-free. In general, training-based method can be more effective**
>
> **A.** We agree that DynamiCrafter and Generative Inbetweening are training-based and fine-tuning-based approaches, respectively. Nonetheless, our method achieves superior video interpolation performance despite its training-free manner, as demonstrated in Table 1.
> More specifically, due to the efficiency of our training-free approach, our method significantly outperforms other SVD-based methods in inference time. Specifically, compared to Generative Inbetweening, our method achieves approximately 6× faster inference, as shown in Table 2.
>
> **W2, Q2. Ablation study is suggested to assess the effect of pre-trained video generation models. Will better video generation models result in better performance?**
>
> **A.** In response to the reviewer’s suggestion, we conducted simple ablation study to assess the effect of VDMs. In this study, we employed SVD-XT, a fine-tuned version of the original SVD. To evaluate the impact of video diffusion models (VDMs), we conducted a comparative analysis between SVD-XT and SVD. Please refer to the revised Appendix B.4. As shown in the table, we see the performance increment with better video generation models.
>
>
> | Pexels | FVD ↓ | LPIPS ↓ |
> |----------|----------|----------|
> | SVD | 608.07 | 0.1577  |
> | SVD-XT (Ours) | **452.34**  | **0.1366**  |
>
> **W3, Q3. Discussion on the limitation and future work.**
>
> **A.** Per the reviewer’s suggestion, we have outlined the limitations and future works in the revised Appendix B.5. As discussed in response to Q1 from Reviewer VMQD, we have not considered the incorporation of text conditioning, as our method is based on SVD. However, when bidirectional sampling is applied to other I2V diffusion models, such as DynamiCrafter, we believe that it becomes feasible to guide actions based on text conditions. Extending our method to control additional text condition is a promising direction, and we are actively considering this for future work.

---

> > ### Author Response · Authors · 2024-11-25
> >
> > Thank you for the constructive comments and raising the score!

---

### Official Review · Reviewer_VMQD · 2024-11-08

**Soundness:** 3
**Presentation:** 3
**Contribution:** 3
**Rating:** 8
**Confidence:** 5

**Summary:**

This work introduces a novel, bidirectional sampling strategy to address off-manifold issues without requiring extensive re-noising or fine-tuning. The proposed method employs sequential sampling along both forward and backward paths, conditioned on the start and end frames, respectively, ensuring more coherent and on-manifold generation of intermediate frames.

**Strengths:**

- Unlike previous fusing strategies, which compute two conditioned outputs in parallel and then fuse them, the proposed bidirectional diffusion sampling strategy samples two conditioned outputs sequentially, which mitigates the off-manifold issue.
- The proposed method can further work with advanced on-manifold guidance techniques (CFG++, DDS) to produce more reliable interpolation results that have better alignment of the input frames.
- This is a training-free method which can efficiently interpolate between two keyframes to generate a 25-frame video at 1024×576 resolution in just 195 seconds on a single 3090 GPU.

**Weaknesses:**

- The experimental results presented lack PSNR scores. While accurate recovery of intermediate frames may not be the primary objective of this study, and high PSNR values might not necessarily indicate superior human preferences, it is still beneficial to include the score to provide readers with a clearer understanding of the work’s potential.
- The proposed bidirectional sampling addresses the off-manifold issue, but it comes with the potential for deviation from the start or end frame, depending on the order of sequential sampling. To mitigate this issue, the authors have incorporated previous guidance techniques. However, I would appreciate a more comprehensive video comparison in the ablation study, in addition to the provided static comparison on a single scene.

**Questions:**

- How to incorporate text conditioning into your framework so users can control the action in the generated frames?
- The results and presentation are sound, except for some minor issues (see weaknesses). I will increase my score once the authors address my concerns.

---

> ### Author Response · Authors · 2024-11-20
> **Reply to reviewer VMQD**
>
> **W1. The experimental results presented lack PSNR scores.**
>
> **A.** In response to the reviewer's request, we additionally provide PSNR scores of our method and baselines. Please refer to the revised Appendix B.1. As shown in the table, our model demonstrates superior performance in both PSNR and SSIM across the DAVIS and Pexels datasets.
>
>
> | DAVIS | PSNR ↑ | SSIM ↑ |
> |----------|----------|----------|
> | TRF  | 16.4078 | 0.4040 |
> | Generative Inbetweening | 16.0377  | 0.4255  |
> | Ours | **17.5215**  | **0.4387**  |
>
>
> | Pexels | PSNR ↑ | SSIM ↑ |
> |----------|----------|----------|
> | TRF  | 19.2670 | 0.5346 |
> | Generative Inbetweening | 20.8624  | **0.6334**  |
> | Ours | **21.4235**  | 0.5939  |
>
> **W2. I would appreciate a more comprehensive video comparison in the ablation study, in addition to the provided static comparison on a single scene.**
>
> **A.** Per the reviewer’s suggestion, we included more comprehensive video comparison of the ablation study in our project page. Please refer to the additional project page for more comprehensive visualization (https://vibid.github.io/supple), and Appendix B.2 for evaluation.
>
> Additionally, more comprehensive video comparisons are available in the second section of our anonymous webpage. The goldfish positioned on the far left in the vanilla TRF model disappears in the intermediate frames. In contrast, the fish exhibits natural movement in the videos generated by our vanilla model. Additionally, while the camel's shape appears blurred and its movement lacks smoothness in videos produced by the vanilla TRF model, our vanilla model demonstrates improved clarity in the camel's form. Furthermore, when advanced guidance techniques are applied, the videos display enhanced quality in the camel's leg shapes and movements.
>
> **Q1. How to incorporate text conditioning into your framework so users can control the action in the generated frames?**
>
> **A.** Currently, our method employs Stable Video Diffusion (SVD), which replaces text embeddings with the CLIP image embedding of the conditioning image. Thereby, the incorporation of text conditioning is limited to control video content through text conditioning. However, when bidirectional sampling is applied to other I2V diffusion models that originally utilize text conditioning, such as DynamiCrafter, it becomes feasible to guide actions based on text conditions. Extending our method to control additional text condition is a promising direction, and we are actively considering this for future work. We have detailed the future work in the revised Appendix B.5.

---

> ### Comment · Reviewer_VMQD · 2024-11-21
>
> Could you provide the scores of FILM? If the video generative models are not competitive with traditional frame interpolation methods, it's totally understandable. You may just have to make it clear in your paper that your method is not designed for accurate reconstruction. If you can modify your paper accordingly to make readers easily understand the difference, I am more than happy to increase my rating to encourage this kind of generative works with amazing visual results.

---

> ### Author Response · Authors · 2024-11-22
>
> Thank you again for the encouraging comments. In response to the reviewer's suggestion, we have included the PSNR and SSIM scores for FILM in the revised Appendix B.1.
>
> As shown in the table, FILM achieves the highest PSNR and SSIM scores compared to all video generative models. This is because FILM is trained using PSNR-oriented losses, such as the L1 loss computed between the ground truth and output frames, which prioritizes pixel-wise reconstruction accuracy. In contrast, our method is specifically designed to generate perceptually natural and smooth motion between consecutive frames, rather than focusing on precise pixel-wise accuracy.
>
> Despite the FLIM's numerical advantages of PSNR and SSIM, it is evident that FILM fails to predict dynamic motion between two bounding frames, as demonstrated in the videos available on our webpage (https://vibid.github.io/). For example, in the cases of a camel and a dog, FILM distorts the appearance of these subjects. This issue arises due to its reliance on PSNR-oriented losses and a network architecture based on an optical flow estimation module, which may compromise the perceptual quality.  This is also confirmed in LIPS and FID score in the following table. To clarify these distinctions, we have revised the paper to ensure readers can better understand the differences.
>
> | | LPIPS ↓ | FID ↓ | FVD ↓ | PSNR ↑ | SSIM ↑ |
> |----------|----------|----------|----------|----------|----------|
> | FILM | 0.2697 | 40.241 | 622.16 | **22.5521** | **0.5346** |
> | TRF  | 0.3102 | 60.278 | 622.15 | 16.4078 | 0.4040 |
> | Generative Inbetweening | 0.2823 | 36.273 | 490.34 | 16.0377  | 0.4255  |
> | Ours | **0.2355** | **35.659** | **399.15** | 17.5215  | 0.4387  |

---

> > ### Comment · Reviewer_VMQD · 2024-11-24
> >
> > Thank you for all your efforts. I’ve increased my rating. Good work!

---

> > > ### Author Response · Authors · 2024-11-25
> > >
> > > Thanks for your very positive feedback and increasing the rating!

---

### Author Response · Authors · 2024-11-20
**General response**

We would like to thank the reviewers for their constructive and thorough reviews.

We are encouraged by the positive reception, as highlighted by the reviewers: **the soundness of our results and presentation** (VMQD), **the convenient, training-free approach of our method** (wFk9), **the impressive results compared to other baselines** (hjqi), and **the simplicity and efficiency of our methodology** (hsxN).


We have made the following major revisions to address the concerns raised from the reviewers.

**1. Additional ablation studies on the guidance methods, CFG++ and DDS.**

In response to the reviewers' suggestions to validate the effectiveness of bidirectional sampling, we conducted ablation studies by incorporating CFG++ and DDS to various baseline models. Please refer to the anonymous webpage (https://vibid.github.io/supple) and the revised Appendix B.2 for experimental details. As shown in the webpage, our vanilla bidirectional sampling shows superior perceptual quality compared to TRF. Furthermore, as shown in the revised Appendix B.2, the videos generated by our proposed method consistently outperform all baseline methods with CFG++ and DDS. The results demonstrate the effectiveness of bidirectional sampling in video interpolation.

**2. More comprehensive video comparisons.**

In response to the suggestions from reviewers VMQD and hjqi, we included more visualization results regarding ablation study on the building blocks. The videos are available at the anonymous webpage (https://vibid.github.io/supple). The roles of each building blocks can be verified from the visualizations and Appendix B.3.

For point-to-point response, please refer to below.

---

### Meta-Review · Area_Chair_49Ft · 2024-12-18

**Metareview:**

Summary: Proposes a frame interpolation method that introduces a novel bi-directional sampling approach, that avoids off-manifold issues, in a training-free manner. It sequentially samples along the forward and backward paths, each conditioning on the start and end frames. Additionally, manifold guidance such as DDS and CFG++ are employed to synergically enhance interpolation results.

Strength:  The bi-directional sampling strategy is novel, and it’s also simple and shown to be effective in qualitative results. Its training-free nature makes it an alternative approach to high-quality frame interpolation without resorting to expensive training or fine-tuning approaches. The paper is written very well, the problem is well motivated, which makes it easy to follow design choices and experimental results. Authors present extensive experimental comparisons that demonstrate the effectiveness of their approach.

Weakness: It currently is not controllable via text. However, authors have identified this as a future work.

Acceptance Reason: See strength. It is a novel approach that shows strong results using a simple and efficient technique. Future works and limitations sections outline a promising avenue for future work.

**Additional Comments On Reviewer Discussion:**

The paper received 2x accept and 2x marginally above acceptance threshold. Reviewers raised a number of points. There is a healthy discussion between the authors and reviewers. Almost all concerts of reviewers are addressed by the authors as acknowledged by the reviewers. There is a singular concern raised by (hsxN and hiqi) on whether the gain in quality is primarily due to the proposed bi-directional sampling or the use of advanced guidance techniques like CFG++ and DDS. I am satisfied by the author's response to this question. These advanced guidance techniques are designed for on-manifold sampling, which makes their method synergically enhanced with these guidance techniques for further improvement.

---

### Decision · Program_Chairs · 2025-01-22

Accept (Poster)